# Enhanced Air Quality Prediction Using a Coupled DVMD Informer-CNN-LSTM Model Optimized with Dung Beetle Algorithm

**DOI:** 10.3390/e26070534

**Published:** 2024-06-21

**Authors:** Yang Wu, Chonghui Qian, Hengjun Huang

**Affiliations:** 1School of Statistics and Data Science, Lanzhou University of Finance and Economics, Lanzhou 730020, China; wuy@lzufe.edu.cn (Y.W.); qianch@lzufe.edu.cn (C.Q.); 2Key Laboratory of Digital Economy and Social Computing Science of Gansu, Lanzhou 730020, China

**Keywords:** air quality index (AQI), Variational Mode Decomposition, dung beetle optimization, Informer, Convolutional Neural Network-Long Short Term Memory

## Abstract

Accurate prediction of air quality is crucial for assessing the state of the atmospheric environment, especially considering the nonlinearity, volatility, and abrupt changes in air quality data. This paper introduces an air quality index (AQI) prediction model based on the Dung Beetle Algorithm (DBO) aimed at overcoming limitations in traditional prediction models, such as inadequate access to data features, challenges in parameter setting, and accuracy constraints. The proposed model optimizes the parameters of Variational Mode Decomposition (VMD) and integrates the Informer adaptive sequential prediction model with the Convolutional Neural Network-Long Short Term Memory (CNN-LSTM). Initially, the correlation coefficient method is utilized to identify key impact features from multivariate weather and meteorological data. Subsequently, penalty factors and the number of variational modes in the VMD are optimized using DBO. The optimized parameters are utilized to develop a variationally constrained model to decompose the air quality sequence. The data are categorized based on approximate entropy, and high-frequency data are fed into the Informer model, while low-frequency data are fed into the CNN-LSTM model. The predicted values of the subsystems are then combined and reconstructed to obtain the AQI prediction results. Evaluation using actual monitoring data from Beijing demonstrates that the proposed coupling prediction model of the air quality index in this paper is superior to other parameter optimization models. The Mean Absolute Error (MAE) decreases by 13.59%, the Root-Mean-Square Error (RMSE) decreases by 7.04%, and the R-square (R^2^) increases by 1.39%. This model surpasses 11 other models in terms of lower error rates and enhances prediction accuracy. Compared with the mainstream swarm intelligence optimization algorithm, DBO, as an optimization algorithm, demonstrates higher computational efficiency and is closer to the actual value. The proposed coupling model provides a new method for air quality index prediction.

## 1. Introduction

As China’s economy grows steadily and industrialization increases, the air pollutants emitted by certain cities have exceeded the tolerable capacity of the environment, leading to a deterioration in ambient air quality. Air pollution has become a serious challenge to social development and ecosystem health. Fine particulate matter (PM2.5), respirable particulate matter (PM10), carbon monoxide (CO), nitrogen dioxide (NO_2_), ozone (O_3_), and sulfur dioxide (SO_2_) are six major air pollutants [1]. Their excessive emissions pose a serious threat to human health, triggering a wide range of diseases affecting the cardiovascular, cerebral, and respiratory systems and causing persistent damage to the human body [2].

According to the Atmospheric Environment Quality Standards (GB3095-2012) set by the Ministry of Ecology and Environment of China, the AQI converts advanced air pollutant quality data into a number, and the concentration of air pollutants in major urban areas exceeds regulatory limits [3]. The AQI ranges from 0 to 500, with 0 representing normal air quality and 500 being the most hazardous air quality. The AQI classifies air quality into six main levels of different health impacts, providing key information about the state of air quality to the public by reflecting the potential impact of different pollution levels on health. Air quality is influenced by various factors. Changes in past pollutant emission intensity and meteorological conditions provide important reference data for predicting future air quality [4]. Meteorological conditions play a crucial role, both directly and indirectly, in the emission, diffusion, formation, and deposition of atmospheric pollutants and are the main driving factors for regional air pollution changes [5]. Therefore, it is crucial to improve the stability and accuracy of air quality prediction caused by the complex problems of nonlinearity and non-stationarity in the AQI due to factors such as meteorological conditions and pollutant type [6]. 

Research on air quality mainly includes numerical simulation methods, statistical modeling theory, and artificial intelligence methods [7,8,9]. Numerical simulation relies on the physical and chemical processes of the atmosphere, combined with meteorological principles and mathematical methods, to simulate large-scale air quality in both horizontal and vertical dimensions. Statistical modeling theory is based on the assumption of data distribution, aimed at explaining causal relationships and emphasizing parameter inference [10]. Machine learning, especially deep learning methods in artificial intelligence, can learn parameters through algorithmic formulas based on input characteristics, identify trends and features in large amounts of data, and make predictive inferences. Taking into account the characteristics of atmospheric pollutants and the influence of various factors such as weather, it has partially addressed the problems of nonlinear fitting, non-stationarity, and complexity in air quality time series, demonstrating superior predictive performance [11]. Traditional machine learning techniques such as Random Forest (RF) [12] and Extreme Gradient Booster (XGBoost) [13] have been widely applied in the field of air quality prediction. Due to the advantages of deep learning in feature extraction and mining, more and more research is focusing on recursive neural networks and their variant models such as Long Short-Term Memory Networks (LSTMs) [14], Gated Recurrent Units (GRUs) [15], Bidirectional Long Short-Term Memory Networks (Bi-LSTMs) [16], Bidirectional Gated Recurrent Units (Bi-GRUs) [17], and Convolutional Long Short-Term Memory Networks (CNN-LSTMs) [18]. Seng et al. [14] proposed a multi-index comprehensive prediction model based on an LSTM, using data on the concentration of major pollutants at representative sites in Beijing, and demonstrated its effectiveness in air quality prediction. Athira et al. [19] used the AirNet dataset of meteorological characteristics and air quality to predict and evaluate atmospheric environmental quality using RNN, LSTM, and GRU models. Compared with traditional models such as ARIMA, they have stronger advantages in terms of evaluation indicators such as R^2^, MAE, and RMSE. Prihatno et al. [16] proposed a multi-density layer model based on a Bi-LSTM to more accurately predict PM2.5 concentration levels in air quality pollutants. Zhang et al. [17] proposed a Bi-GRU model based on spatial geographic location information. The combination of historical air quality monitoring data and meteorological monitoring data has improved the prediction accuracy of recurrent neural networks in air quality. However, a single predictive model may be limited by specific algorithms or model structures and may not be able to adapt well to complex data patterns or relationships. In addition, when facing multi-source data, it may not be possible to capture complex patterns in the data, and the data may be incomplete, making it difficult to fully utilize information. Therefore, how to improve the accuracy of prediction methods has become a key issue in current air quality prediction research.

A Convolutional Neural Network (CNN) can capture uneven changes in air pollution data, effectively extract data features, expand the scope of analysis, and evaluate the connectivity between features. The memory units in an LSTM make neural networks more suitable for time series analysis and modeling. The CNN-LSTM model combines the advantages of both methods and adaptively extracts features layer by layer, enhancing the network’s learning ability and more effectively extracting information from time series data. Yan et al. [18] combined the advantages of CNNs and LSTMs to establish a CNN-LSTM coupled prediction model based on multi-temporal seasonal station data from Beijing. Compared with traditional single prediction models, it significantly reduces AQI errors. To further improve the accuracy of AQI prediction, Zhang et al. [20] utilized the effective feature extraction function of a CNN for data feature extraction, constructed a feature vector sequence, and fed it into the LSTM network to learn the variation patterns of air quality data, demonstrating the performance advantages of the CNN-LSTM model. Additionally, attention mechanisms have greatly improved the computational efficiency and accuracy of models in feature extraction and parallel computing. They can also capture the time dependence and correlation between long-term data indicators. In recent years, they have been widely used by scholars. Long et al. [21] proposed an improved Informer [22] model based on the Transformer [23] architecture. The Informer model, which improves the self-attention mechanism, has strong advantages in feature extraction and parallel decoding, enhancing the accuracy of air quality prediction.

In addition to the rapid development of artificial intelligence-related prediction models, the effective preprocessing of datasets plays an important role in extracting data features, reducing the complexity of time series data and modeling, and improving prediction accuracy. The application of the modal function helps to deeply analyze and reveal the periodicity, trend, and abnormal changes of air quality series, and signal decomposition methods can obtain more accurate prediction accuracy [24]. Huang et al. [25] proposed an Improved Particle Swarm Optimization algorithm (IPSO) based on the Empirical Mode Decomposition (EMD) method of the LSTM to optimize the parameters of the LSTM for air quality modeling. However, to address the issues of mode mixing and end effects in EMD and other decomposition methods, scholars have gradually developed and applied the VMD [26]. VMD is a non-recursive modal signal processing technique that decomposes time series into multiple relatively smooth subsequences with different frequency scales, which can reduce the non-stationarity and complexity of time series. VMD is highly robust to noise and can effectively overcome mode mixing and achieve suitable decomposition results. Wu et al. [27] proposed a VMD-LSTM mixed air quality prediction model, which utilizes VMD to decompose the original AQI sequence into subsequences with different frequencies and rearranges these subsequences using sample entropy, addressing issues of over-decomposition and a large amount of calculation.

Data preprocessing techniques may encounter challenges related to local optimization and slow convergence during post-processing. Moreover, improper parameter selection can degrade the quality of decomposition results, thereby impacting the accuracy of air quality prediction models [28]. The swarm intelligence optimization algorithm has a strong ability to seek optimality by imitating the search behavior of population predation in nature and is frequently employed to optimize model hyperparameters. By integrating the swarm intelligence optimization algorithm with signal decomposition techniques such as VMD, the challenge of hyperparameter setting is effectively mitigated, and the performance of the integrated model is enhanced [29]. Shao et al. [30] introduced an adaptive parameter search technique for VMD based on the DBO algorithm to optimize VMD parameters, which exhibits a fast iteration rate and yields good optimization effects. Additionally, they employed the Informer and XGBoost methods to handle high-frequency and low-frequency feature data of approximate entropy classification and ultimately superimposed the reconstructed predicted values to achieve high prediction accuracy. Wang et al. [31] utilized the multi-objective Gray Wolf Optimization (GWO) algorithm [32] in conjunction with the VMD method to effectively extract the intrinsic information of PM2.5 data, thereby reducing the computational complexity of the model and enabling timely detection and warning of air pollution. Chang et al. [33] merged wavelet decomposition [34] and the Particle Swarm algorithm (PSO) [35] to introduce a hybrid framework model for predicting the accuracy of major air pollutants. Liu et al. [36] employed the Sparrow Search Algorithm (SSA) [37] to optimize LSTM parameters, resulting in improved prediction performance compared with that of a standalone neural network model in terms of MAPE, RMSE, MAE, and R^2^ metrics. Xu et al. [38] utilized the Whale Optimization Algorithm (WOA) [39] to optimize SVM parameters, leading to the development of a hybrid ICEEMD-WOA-SVM warning model, which was empirically validated in the cities of Taiyuan, Harbin, and Chongqing. Subsequent developments in algorithmic models should integrate multiple algorithms to enhance prediction accuracy and generalization while capturing data features.

Current research on air quality forecasting encounters several challenges. Firstly, the nonlinearity, complexity, and nonsmoothness characteristics of the air quality series make it challenging for a single feature to capture the overall changes. Various influencing factors, such as common atmospheric pollutants and meteorology, should be considered in the prediction of the AQI [40]. Secondly, some commonly used original data decomposition methods, such as VMD, often rely on trial-and-error methods based on subjective experience, making it difficult to find appropriate parameters, which may reduce the quality of decomposition results due to improper parameter selection, thus affecting the prediction accuracy of the model [41]. Additionally, while most optimization algorithms are utilized for refining prediction models, significant potential remains for data preprocessing. Due to the complexity and instability of the time series data of the main air quality influencing factors, data preprocessing techniques, although generally capable of mining the potential characteristics of air pollutants to improve the prediction accuracy [42], suffer from the issues of falling into local optima and slow convergence [43]. Based on the above issues, this paper proposes a coupled DVMD-Informer-CNN-LSTM model based on the optimization of VMD parameters by DBO, which combines Informer and CNN-LSTM models for application in air quality prediction. The model uses the correlation coefficient method to select the main influential features from meteorological data and atmospheric pollutants, with VMD optimized by the improved Dung Beetle Algorithm (DBO). Envelope entropy is used as the fitness value to determine the penalty factor and the number of Intrinsic Mode Function (IMF) decomposition layers for the variational mode fraction. Compared with the aforementioned GWO, PSO, SSA and WOA, DBO shows significant competitive performance in terms of both accuracy and stability of the solution accuracy and realizes the accurate decomposition of complex data. Compared with the relevant benchmark models set in this paper, the proposed DVMD-Informer-CNN-LSTM air quality prediction model is innovative and accuracy-enhancing in terms of parameter optimization, model architecture, and prediction performance. The contributions of this paper can be summarized as follows:

(1) To effectively capture and process complex features of air quality data and enhance AQI prediction accuracy. A DVMD-Informer-CNN-LSTM multi-factor coupled integration framework model based on DBO, VMD, Informer, and CNN-LSTM is proposed. This model can better capture the long and short-term dependencies in the time series through the improved Informer algorithm, which includes a long- and short-term global self-attention mechanism, as well as the CNN-LSTM algorithm to enhance the model’s ability to process sequence data. The Informer model is specifically chosen for its efficiency in handling high-frequency data, which often contains complex patterns and significant noise. Informer’s probabilistic sparse self-attention mechanism reduces computational complexity while maintaining robust performance, making it highly suitable for high-frequency datasets. Secondly, a multi-level feature extraction strategy is employed to process the original time series data through various feature extraction modules. This strategy significantly reduces the prediction error stemming from the dataset’s complexity by extracting features at different levels from both high- and low-frequency data following the information entropy decomposition. The CNN-LSTM model is utilized for low-frequency data due to its ability to capture features and temporal dependencies through CNN and LSTM layers. This combination is particularly effective for low-frequency data. Finally, the parallel decoder structure enables the distributed processing of different prediction tasks simultaneously, enhancing the parallelism and efficiency of the model. This accelerates the training and inference speed of the model while maintaining high prediction accuracy, demonstrating the model’s practical value. 

(2) A DBO-based optimization algorithm is employed to adjust the adaptive parameters of the VMD. The DBO algorithm, known for its rapid iteration rate and effective optimization, optimizes the penalty factor and the number of decomposition layers of the VMD. Envelope entropy is used as the adaptive value, while information entropy serves as the boundary between high-frequency and low-frequency data. This method prevents modal mixing and pseudo-componentization during VMD decomposition, enabling more precise parameter identification. It minimizes errors due to subjective parameter adjustments and integrates model fusion to significantly enhance prediction accuracy.

(3) Using 26,110 data points from 13 major air quality monitoring stations in Beijing from 2021 to 2023, the proposed DVMD-Informer-CNN-LSTM air quality multi-factor coupled comprehensive prediction model and 11 benchmark models were applied to improve the accuracy, stability, and interpretability of the model through data preprocessing. The evaluation metrics, including MAE, RMSE, and $R^{2}$, demonstrate the accuracy and adaptability of the model.

The remainder of the paper is organized as follows: Section 2 outlines the algorithmic principles and detailed implementation steps of the DVMD-Informer-CNN-LSTM coupled model. Section 3 concentrates on presenting the numerical experimental results required for the model. This section includes discussions on the data sources and preprocessing methods, correlation analysis of key factors affecting air quality, parameter optimization and data decomposition of the VMD by DBO, standardization of the experimental setup, and accuracy assessment of the results. Section 4 summarizes the advantages and limitations of the proposed model and offers an overview of future research prospects.

## 2. Models and Algorithms 

### 2.1. Variational Modal Decomposition (VMD)

Given the highly nonlinear and nonsmooth uncertainty inherent in AQI, efficient dataset preprocessing can enhance data feature extraction and diminish time series data and modeling complexity, thereby enhancing prediction accuracy.

Several signal decomposition techniques have been successfully employed in time series prediction, including Discrete Wavelet Transform (DWT) [44], EMD [26], and Ensemble Empirical Mode Decomposition (EWT) [45]. While DWT is effective in capturing signal changes at different scales, it relies on predefined wavelet bases, which may not adapt well to the intrinsic characteristics of complex signals, potentially leading to suboptimal decomposition for highly irregular data. EMD is a widely used technique that decomposes a signal into Intrinsic Mode Functions (IMFs). However, it often suffers from mode mixing and end effects, which can compromise the accuracy and reliability of the decomposition. EWT was developed to mitigate some of these issues but remains computationally intensive and susceptible to reconstruction errors.

VMD offers significant advantages over DWT, EMD, and EWT. Unlike DWT, VMD does not require predefined basis functions and instead uses adaptive basis functions that better fit the data’s inherent properties. VMD effectively addresses the mode mixing and end effects associated with EMD by iteratively searching for modes with minimal interference and distinct frequency bands. This results in more precise and stable decomposition outcomes, which are crucial for accurate AQI prediction. Moreover, VMD’s robustness to noise ensures reliable performance even with noisy datasets, enhancing the overall prediction accuracy. Therefore, we chose VMD for its ability to adaptively decompose complex signals, handle noise effectively, and produce stable and accurate components suitable for subsequent predictive modeling.

VMD is a method grounded in variational methods that decomposes nonlinear and nonsmooth signals into multiple IMFs [26]. Fundamentally, VMD aims to minimize interference between each IMF and other frequency bands via iterative optimization. This prevents information overlap and enhances robustness to noise and interference. In comparison to other techniques, VMD demonstrates superior accuracy, faster computation speed, and better noise immunity when handling data center frequency components.
(1)$$\min_{\left\{u_{k}\},\{\omega_{k}\right\}}\left\{{\sum_{k}\left\|\partial_{t}\left[\left(\delta (t)+\frac{j}{\pi t}\right)⊗u_{k}(t)\right]e^{-j\omega_{k}t}\right\|}_{2}^{2}\right\}$$
(2)$$s.t.\sum_{k=1}^{K}u_{k}=f(t)$$
where $\left\{u_{k}\right\}$ and $\left\{\omega_{k}\right\}$ denote the modal components and center frequency; δ(t) is the Dirac function; f is the original signal; and ⊗ is used for the convolution operation.

Solving variational constrained problems, penalty factor α and Lagrange multiplier λ are introduced to transform the constrained variational problem into an unconstrained variational problem. The augmented Lagrange expression is derived as:(3)$$L=(\{u_{k}\},\{\omega_{k}\},\lambda )=\alpha \sum_{k=1}^{K}\left|\right|\partial_{t}\left[\left(\delta (t)+j/\pi t)⊗u_{k}(t\right)\right]e^{-j\omega_{k}t}{\left|\right|}_{2}^{2}\;+\left|\right|f(t)-\sum_{k=1}^{K}u_{k}(t){\left|\right|}_{2}^{2}+\left[\lambda (t),f(t)-\sum_{k=1}^{K}u_{k}(t)\right]$$

The modal and center frequencies are updated using the multiplier alternating direction method to seek the saddle points of the augmented Lagrangian function, $\left\{u_{k}\right\}$, $\left\{\omega_{k}\right\}$ and λ with the following expressions:(4)$${\hat{u}}_{k}^{n+1}(\omega )\leftarrow \frac{f(\omega )-\sum_{i\neq k}{\hat{u}}_{i}(\omega )+\frac{\hat{\lambda }(\omega )}{2}}{1+2\alpha {\left(\omega -\omega_{k}\right)}^{2}}$$
(5)$$\omega_{k}^{n+1}\leftarrow \frac{\int_{0}^{\infty }\omega {\left|u_{k}^{n+1}(\omega )\right|}^{2}d\omega }{\int_{0}^{\infty }{\left|u_{k}^{n+1}(\omega )\right|}^{2}d\omega }$$
(6)$$\left.{\hat{\lambda }}^{n+1}(\omega )\leftarrow {\hat{\lambda }}^{n}(\omega )+\gamma \left(f(\omega )-\sum_{k}{\hat{u}}_{k}^{n+1}(\omega )\right.\right)$$
where γ is the noise tolerance, n is the number of iterations, and the Fourier transforms correspond to $u_{k}^{n+1}(\omega )$, $u_{i}(\omega )$, f(ω), and $\hat{\lambda }(\omega )$ are $u_{k}^{n+1}(t)$, $u_{i}(t)$, f(t), and λ(ω).

### 2.2. DBO-Optimized VMD (DVMD)

Unlike existing optimization algorithms such as GWO, PSO, SSA, and WOA, among others, the Dung Beetle Optimization Algorithm (DBO) considers both global exploration and local exploitation while exhibiting fast convergence speed and high solution accuracy. The optimization technique of adaptive parameter finding can better optimize the penalty factor and the number of intrinsic modes of VMD, and through the dung beetle’s positional changes, the global optimal position and the adaptation value can be obtained, thereby facilitating accurate AQI decomposition prediction.

The population-based DBO algorithm, proposed by Xue in November 2022 [46], draws inspiration from the dung beetle’s behaviors of rolling, dancing, foraging, stealing, and reproducing. To simulate the rolling behavior, the dung beetle is required to move in a specified direction throughout the search space.
(7)$$x_{i}(t+1)=x_{i}(t)+\alpha kx_{i}(t-1)+b\;\Delta \;x$$
(8)$$\Delta x=\;|x_{i}(t)-X^{w}|$$
where t denotes the current iteration number, $x_{i}(t)$ denotes the information of dung beetle i at the tth iteration, k ranges from 0 to 0.2, and $X^{w}$ denotes the deflection coefficient. Δx denotes the worst-case global position and denotes the change in light intensity.

When the dung beetle is prevented from moving forward by an obstacle, a dance behavior repositioning is required, and updating the rolling ball dung beetle position can be expressed as follows:(9)$$x_{i}(t+1)=x_{i}(t)+\tan(\theta )|x_{i}(t)-x_{i}(t-1)|$$
where $x_{i}(t)-x_{i}(t-1)$ denotes the difference between the position of the i dung beetle at the tth iteration and its position at the t−1th iteration, and θ is the angle of deflection belonging to 0 to θ. The position of the dung beetle is not updated when θ=0, π/2, and π.

Dung beetles will hide dung balls for spawning, and a boundary selection strategy is needed to determine the spawning boundary to simulate the spawning region, which can be expressed as follows:(10)$$Lb^{*}=\max(X^{*}(1-R),Lb)$$
(11)$$Ub^{*}=\min(X^{*}(1+R),Ub)$$
(12)$$B_{i}(t+1)=X^{*}+b_{1}(B_{i}(t)-Lb^{*})+b_{2}(B_{i}(t)-Ub^{*})$$
where $X^{*}$ denotes the current local optimal position, $Lb^{*}$ and $Ub^{*}$ denote the lower and upper limits of the spawning area, respectively, $R=1-t/T_{\max}$, $T_{\max}$ denote the maximum number of iterations, and Lb and Ub denote the lower and upper limits of the optimization. $B_{i}(t)$ is the position information of the ith female sphere at the tth iteration, denotes two independent random vectors of size 1×D, and D denotes the dimension of the optimization problem.

Dung beetles have a foraging behavior that searches for physical objects, and the optimal foraging boundary can be expressed as follows:(13)$$Lb^{b}=\max(X^{b}(1-R),Lb)$$
(14)$$Ub^{b}=\min(X^{b}(1+R),Ub)$$
(15)$$x_{i}(t+1)=x_{i}(t)+C_{1}(x_{i}(t)-Lb^{b})+C_{2}(x_{i}(t)-Ub^{b})$$
where $X^{b}$ is the global optimal location, $Lb^{b}$ and $Ub^{b}$ denote the lower and upper bounds of the optimal foraging area, respectively, $C_{1}$ denotes a random number, and $C_{2}$ denotes a random vector belonging to (0, 1).

Some dung beetles have the behavior of stealing dung balls from other dung beetles, and the update of the location of the area of theft can be expressed as follows:(16)$$\;x_{i}(t+1)=X^{b}+S\times g(|x_{i}(t)-X^{*}+x_{i}(t)-X^{b}|)$$
where g is a normally distributed random vector of size 1×D and S is a constant value.

The dung beetle population contains N individuals, and each individual i represents a candidate solution. At the tth iteration, the position vector of the ith agent is represented by 1, where D is the dimension of the search space, and the allocation ratio is not specified and can be set according to the actual application problem. In Figure 1, the size of the dung beetle population was chosen to be 30. Green, yellow, blue, and red rectangles represent the number of dung beetles rolling, laying eggs, foraging, and stealing, with numbers 6, 6, 7, and 11, respectively. The sum of the numbers corresponds to the population size, and the percentages are about 20%, 20%, 25% and 35%.

To summarize, the DBO algorithm, as a novel optimization technique, confronts an optimization problem comprising six main steps: initializing the dung beetle population and the parameters of the DBO algorithm; calculating the fitness values of all the individuals according to the objective function; updating the positions of all the dung beetles; determining whether each individual crosses the boundary; updating the current optimal solution and its fitness value; iteratively performing the aforementioned steps until the termination criterion is met; and finally outputting the global optimal solution and its corresponding fitness value (Figure 2). In the aforementioned model, the number of decomposition layers (k) and the penalty factor parameter (α) of VMD are optimized using DBO. The number of modes k and the penalty factor α have the most significant influence on the VMD decomposition outcomes. Elevated values of k may result in an overdecomposition of the original signal, yielding spurious components. Conversely, lower values of k may result in the underdecomposition of the signal, thus leading to modal aliasing. The size of the penalty parameter determines the size of the bandwidth of each component; a larger value will make the modal function narrower, resulting in loss of bandwidth information, while a smaller value will carry too much interference information, resulting in redundancy. Determining the inherent relationship between these two parameters, optimizing, and validating the appropriate [k, α] values constitute the crucial aspect of VMD decomposition. Envelope entropy can more effectively reflect the sparse characteristics of the initial signal: when the modal component contains less feature information, the envelope entropy is higher; conversely, when there is less noise and more feature information in the IMF, the envelope entropy is lower. This property can be leveraged as the target fitness function for DBO optimization, as depicted below:(17)$$E_{p}=-\overset{m}{\underset{j=1}{\sum^{\;}}}p_{j}\mathrm{lg}p_{j}$$
(18)$$p_{j}=b(j)/\overset{m}{\underset{j=1}{\sum^{\;}}}b(j)$$

$p_{j}$ is a sequence of probability distributions obtained by normalizing b(j), and b(j) represents the envelope signal of the Hilbert demodulated signal.

The DBO algorithm is employed for global optimization of the two parameters, k and α, of the VMD simultaneously to achieve adaptive decomposition of the air quality sequence. We adopted the optimization of VMD for data decomposition, specifically to manage the inherent complexities of AQI time series data. The detailed process is as follows: First, the time series is decomposed by setting the number of decomposition modes and penalty parameters, yielding multiple mode functions. To prevent data leakage, we meticulously followed standard procedures for time series prediction, ensuring the independence of training and test data. During the decomposition, we referenced the relevant literature [47] and employed cross-validation techniques to ensure that test data were not exposed during model training. In the air quality prediction model presented in this study, the DVMD algorithm is employed to achieve precise decomposition of complex data. In our approach, we employed a static integration method to combine the predictive outputs of the Informer and CNN-LSTM models. The predictions from each model are weighted and averaged to obtain the final AQI prediction. The choice of static integration was based on its simplicity and proven effectiveness in our preliminary tests. Although dynamic integration methods, which adaptively adjust weights based on performance metrics, can potentially offer better accuracy, our experiments indicated that the static integration already provided sufficient predictive precision.

### 2.3. Informer

Due to the attention mechanism in feature extraction, parallel computation has significantly improved the model’s computational efficiency and accuracy, and significant progress has been made this year. The Informer model with an improved self-attention mechanism has a strong advantage in feature extraction and parallel decoding and can capture the temporal dependence and correlation between long-time data indicators, which improves the prediction accuracy of the AQI.

The Informer model was proposed by Zhou et al. [22] in 2020 and introduces an enhanced self-attention mechanism that excels in processing long sequences. High-frequency data, characterized by significant noise and complex patterns, benefits from the Informer model’s ability to efficiently capture long-term dependencies and correlations between data points and can effectively capture the precise long-term correlation coupling between outputs and inputs due to the existence of quadratic time complexity, high memory utilization, and limitations inherent in the architecture of codecs in the traditional Transformer model. The Informer model proposed the ProbeSparse sparse self-attention mechanism, which achieves O(LlogL) in time complexity and memory usage. In addition, self-attention can highlight the dominant attention by halving the cascaded layer inputs and efficiently handling extremely long input sequences, improving the inference speed of long sequences. The Informer model architecture is shown in Figure 3, where the probabilistic sparse self-attention mechanism is used instead of regular self-attention. The encoder on the left receives a large number of green long sequence inputs so that the blue trapezoid is a self-attention extracting operation, and to greatly reduce the network size, the robustness is enhanced by using cascading to extract the main attention. The decoder on the right also receives long sequential inputs, fills the target element with zeros, measures the weighted self-attention composition of the feature map, and generates the orange predicted output element.

The Informer model solves the time complexity problem of self-attentive dot product computation in optimization problems using the probabilistic sparse self-attention algorithm in the encoder. The dot product computation is performed by sampling each point with the equation shown below [48]:(19)$$A(Q,K,V)=softmax(\frac{\bar{Q}K^{T}}{\sqrt{d}})V$$

Q, K, and V are three matrices of the same size obtained by linearly transforming the input feature variables, where $Q\in R^{L_{Q}\times d}$, $K\in R^{L_{K}\times d}$, $V\in R^{L_{V}\times d}$. d are the input dimensions, $\bar{Q}$ are obtained by probabilistic sparsification Q, and Softmax is the activation function.

An extraction operation is used to prioritize superior features with dominant features and produce a focused self-attentive feature map in the next layer. It drastically cuts down the time dimension of the input. The public presentation is as follows:(20)$$X_{j+1}^{t}=Maxpool\left(\;ELU(\;Conv1d({\left[X_{j}^{t}\right]}_{AB})\;)\;\right)$$
where ${\left[X_{j}^{t}\right]}_{AB}$ denotes the multi-head probabilistic sparse self-attention module, Conv1d is a one-dimensional convolutional filter executed in the time dimension using the activation function ELU, and MaxPool is the maximum pooling layer with a step size of 2. After $X^{t}$ downsampling the stacked layers, the memory usage will be reduced to O((2−ϵ)LlogL), with ϵ being a very small value.

The decoder gets a long input sequence and places a zero at the predicted target location. The decoder is passed through the attention layers to produce the expected output. Each decoder layer performs a feature extraction operation based on the input in the direction of the target, a two-layer stack of multi-head attention mechanism level composed decoders that use probabilistic sparse self-attention in the first layer to compute the current output using only a priori data. The multi-head probabilistic sparse self-attention obtains a long input sequence in the second stage and sets zero at the projected target point. The projected output is then generated by masking the attention layers in the last step. Each decoding layer performs feature extraction operations in the target direction based on the given input to complete the decoding process. In the proposed air quality prediction model, Informer is used to predict the high-frequency components of the DBO-optimized VMD.

### 2.4. CNN-LSTM Model

Convolutional Neural Networks have been widely used for time series data analysis due to their powerful feature extraction properties, which capture the temporal dependence present in AQI data by extracting the relationships in the spatial structure of multi-dimensional time series data [49]. Memory units in the LSTMs make the neural networks more suitable for time series analysis and modeling, and the layer-by-layer adaptive feature extraction gives the networks a stronger learning ability to acquire information in time series more efficiently. The CNN-LSTM hybrid model, which can combine the advantages of CNN and LSTM, enables the model to better handle the data of AQI and make predictions.

The network structure of CNNs mainly contains an input layer, a convolutional layer, a pooling layer, a fully connected layer, and an output layer. In addition to this, CNNs share a convolutional kernel in different regions, and the same convolutional kernel has the same weight parameters, which realizes weight sharing. The pooling layer can also be mainly used for feature dimensionality reduction to reduce the number of parameters and decrease the network complexity. Long Short-Term Memory Neural Networks can be a good solution to the long-distance dependency problem. The network structure mainly contains an input layer, an LSTM layer, a fully connected layer, and an output layer [50]. In the LSTM network structure, three gate units are added inside the neurons to control the transmission of information, i.e., input gate, forgetting gate, and output gate, by gating the memory units. The problem of information loss and stability in long-time sequence propagation is solved, and the problems of gradient vanishing and gradient explosion, which exist in recurrent neural networks such as an RNN, are solved. The structure of the constructed CNN-LSTM framework is shown in Figure 4.

In the CNN-LSTM model, the convolutional layer of the CNN is used for extracting local features from time series data. For the pooling layer to further obtain the hidden information and reduce the feature dimensions, the pooled information is used to learn the long-term dependencies in the data through the LSTM network to obtain the features with long-term dependencies. By calculating the loss function to update each hidden layer and selecting a suitable optimization algorithm to improve the prediction of the model, the CNN-LSTM is found to have a higher accuracy rate in AQI prediction compared with other algorithms. In the new model proposed in this paper, the low-frequency components of the DVMD decomposition are predicted using the well-performing CNN-LSTM model.

### 2.5. DVMD-Informer-CNN-LSTM Model

To improve the air quality prediction accuracy, based on the optimization performance of the DVMD algorithm, the DVMD algorithm, Informer algorithm, and CNN-LSTM algorithm are coupled, and the DVMD-Informer-CNN-LSTM combined air quality prediction model is proposed. The selection of DVMD, Informer, and CNN-LSTM as the prediction models is based on a comprehensive consideration of model performance, complexity, computational resources, and diversity. In terms of model performance, these proposed models have shown excellent performance in previous studies, especially in processing time series data with high prediction accuracy. Under limited computing resources, a combination of models that can complete training and prediction in a reasonable time was selected to improve the robustness and generalization ability of the prediction results.

The model consists of three parts: data preprocessing, parameter optimization decomposition, and coupled model prediction and the flow chart is shown in Figure 5. The core of the model uses the DBO algorithm to optimize the decomposition modes and penalty factor parameters in the VMD model, adopts the envelope entropy as the adaptation value, and reflects the high-frequency and low-frequency data values of the approximate entropy decomposition of the tiny fluctuation and nonlinear change characteristics in the time series, which are inputted into the Informer and the CNN-LSTM model for the combined prediction. The new model solves the problems of prediction bias due to data complexity and insufficient feature capture, neglecting the influence of factors other than time series, and poor accuracy due to human intervention that exists in the field of traditional air quality prediction. The specific steps to realize the model are as follows.

Step 1: Data preprocessing. Collect the data of six major air pollutants and key meteorological features as the original dataset, and perform data standardization and linear interpolation to fill in the missing values of the data and remove the outliers, data smoothing, and other preprocessing operations on the data.

Step 2: Data feature selection. The Spearman correlation coefficient method is used to screen the meteorological characteristics and the six major air pollutants data for the importance of their influence characteristics, to determine the key influencing factors and use them as inputs to the dataset, to reduce the complexity of the data, and to provide a basis for the subsequent improvement of the model prediction accuracy.

Step 3: Model parameter optimization. After verifying the efficiency and performance of the DBO algorithm. The DVMD algorithm is applied to optimize the decomposition mode k and penalty factor, and the IMF components are obtained by decomposing the historical AQI data. According to the approximate entropy value of each IMF component, the m IMF components are divided into f low-frequency decomposition data and m-f high-frequency decomposition data, which provide the basis for subsequent model selection and prediction.

Step 4: Feature input and prediction output. The n influential features and the historical data of AQI obtained after feature screening are used as the feature inputs of the model and the prediction outputs of the model. The Informer model is constructed to predict the f low-frequency decomposition data, and the CNN-LSTM prediction model is constructed to predict the m-f high-frequency decomposition parts. Finally, the prediction results of the m components are reconstructed and superimposed to generate the final prediction results.

Step 5: Model evaluation. The prediction ability of the DVMD-Informer-CNN-LSTM model and the proposed 11 comparative models is measured using MAE, RMSE, and R^2^ evaluation metrics. The VMD decomposition method enhanced by the DBO optimization algorithm has been successfully applied to air quality prediction models.

## 3. Experimental Analysis

### 3.1. Data Source

As the center of the capital, Beijing is situated in a warm temperate semi-humid region characterized by a continental monsoon climate. This characteristic is highly representative of urban agglomerations with similar climates. The experimental data were sourced from the weather conditions of Beijing on the World Historical Weather website (https://rp5.ru/) and the real-time national urban air quality release platform of the China Environmental Monitoring General Station (https://air.cnemc.cn:18007/). Air quality data for Beijing, covering the three-year period from 1 January 2021, to 31 December 2023, were selected. This dataset includes the air quality index (AQI) and major pollutants such as PM2.5, PM10, SO_2_, NO_2_, O_3_, and CO. Meteorological data encompass a variety of climatic conditions and air quality indicators, including temperature, barometric pressure, relative humidity, wind speed, and precipitation.

From 2021 to 2023, the AQI in Beijing displayed significant seasonal variations, forming a ‘U’ shaped pattern with higher levels in winter and lower in summer. Data were collected hourly and updated in real-time from thirteen meteorological base stations across various areas of Beijing. Modeling experiments utilizing representative monitoring data from Beijing provide robust evidence of the model’s novelty, applicability, and the validity and credibility of the results. The primary prediction target is the AQI. Initially, the collected experimental data must be preprocessed.

### 3.2. Data Preprocessing

#### 3.2.1. Abnormal Data Smoothing and Standardization

Some nodes in the dataset exhibit abrupt data changes, largely attributed to significant fluctuations in air quality stemming from secondary aerosol formation. These fluctuations are induced by unexpected events such as dust explosions, large-scale human gatherings, and increased soil dust resuspension [28]. Data smoothing is a commonly used data preprocessing technique aimed at reducing noise and eliminating data irregularities to enhance the accuracy and stability of the model, thereby rendering the data smoother and more continuous for modeling and analysis. Assuming that the data are based on normal distribution, univariate anomaly correction is performed on the outlier nodes. A threshold of 99% is set; data points exceeding this threshold are identified as outliers and subsequently removed. After the integration and cleaning of the data, the number of data rows is 26,110 rows. The log1*p* function can address deviations from the Gaussian distribution and outliers in the data. This function was applied to the 26,110 rows of smoothed data.
(21)log1p=log(x+1)
where *x* is the collated data value. 

Linear interpolation was employed to judiciously supplement the original dataset due to the limited extent of missing data, ensuring alignment with the dataset’s original trend. Normalization of the processed data is necessary after performing the normalization operation on the data.

#### 3.2.2. Statistical Tests

To ensure the validity of the dataset, it is necessary to test the stationarity and autocorrelation of time series data before applying predictive modeling. Non-stationary data may lead to misleading and inaccurate predictions. To verify the stationarity of AQI time series data, enhanced Dickey-Fuller (ADF) [51] and Kwiatkowski-Phillips-Schmidt-Shin (KPSS) [52] tests were used. The ADF test is used to estimate the existence of unit roots in a sequence, while the KPSS test is used to evaluate the trend stationarity of the sequence. At the same time, the Ljung-Box Q test [53] was used to detect the autocorrelation of AQI time series data, with a lag order set to 10. The ADF test evaluates the presence of unit roots in time series samples. A unit root represents non-stationarity. The null hypothesis of the ADF test is that the time series has a unit root (non-stationary), while the alternative hypothesis is that the time series is stationary. The KPSS test supplements the ADF test by evaluating the null hypothesis that a time series is stationary around a deterministic trend (stationary), rather than the alternative hypothesis that it is a unit root process (non-stationary). We conducted ADF and KPSS tests on the air quality data, and the results are shown in Table 1. The ADF test results indicate that there are no unit roots in the sequence, while the KPSS test results indicate that the sequence has trend stationarity. The Ljung-Box Q test shows that the *p*-value of 0.0 is less than 0.05, which rejects the null hypothesis of no autocorrelation and indicates that the AQI data in Beijing has strong autocorrelation. The time series data of the Beijing air quality index has stationarity and strong autocorrelation, indicating a strong trend in the data, but there is some predictive noise. Traditional econometric models find it difficult to accurately predict complex air quality index sequences without effective noise reduction and nonlinear processing. It is necessary to decompose the data, reduce data noise, and use coupled model methods to achieve accurate AQI prediction.

#### 3.2.3. Dataset Partitioning

The organized data are divided, of which 70% is the training set, 20% is the testing set and 10% is the validation set.

After data integration, cleaning, smoothing, normalization, and division, the experimental data need to be data analyzed to observe and discover the correlation and other features of the data.

### 3.3. Data Correlation Analysis

Relevant literature [54,55] demonstrates that meteorological variables related to air pollutants like PM2.5, PM10, and weather conditions such as temperature and barometric pressure, can influence AQI changes. Therefore, it is essential to consider the time factor and the impacts of these variables on AQI forecasting. However, as each variable exhibits varying levels of influence, those with minimal or no impact are filtered out using correlation coefficients. Feature selection plays a critical role in the preprocessing of meteorological and pollutant data, aiding in the avoidance of overfitting, enhancement of accuracy, and reduction of computational efforts. Both the Pearson and Spearman correlation coefficient methods are commonly employed to evaluate correlations between two variables. However, the Spearman correlation coefficient method is preferable over the Pearson method, especially when dealing with non-normally distributed variables or ordinal data. Considering the periodic nature and fitting of the data, the Spearman method is more suitable for calculating correlations between features and predictor variables.

In statistics, the Spearman correlation coefficient method employs monotonic equations to assess correlations between two variables. A positive correlation coefficient indicates a similar trend in changes between two variables, whereas a negative one signifies opposing trends. A correlation coefficient of 0 indicates no correlation between the variables. The formula for calculating the correlation coefficient between two variables is as follows:(22)$$\rho (x,y)=\frac{\overset{i=1}{\underset{n}{\sum^{\;}}}(x_{i}-\bar{x})(y_{i}-\bar{y})}{\sqrt{\overset{i=1}{\underset{n}{\sum^{\;}}}{\left(x_{i}-\bar{x}\right)}^{2}{\left(y_{i}-\bar{y}\right)}^{2}}}$$
where ρ is the correlation coefficient between *x* and *y*, *n* is the size of the dataset, and are the mean values, and the heat map of the characteristic correlation coefficients is shown in Figure 5.

Figure 6 shows the correlation coefficients of AQI with other variables, where the magnitude of these coefficients signifies the significance of their effects on AQI. Absolute values exceeding a specific threshold denote a strong correlation with the forecast target. The correlation coefficient matrix reveals that meteorological characteristics are weakly correlated. Therefore, using these weakly correlated variables as predictors for AQI may lead to issues like overfitting, decreased accuracy, and increased complexity in features. The description of each feature depicted in the heatmap is presented in Table 2.

### 3.4. Optimizing Parameter Decomposition Data

To determine the parameter range of DBO. The performance of DBO must be evaluated to illustrate its advantages and reflect its underlying principles. Literature [18] provides the recommended range of values for each DBO parameter, and the performance of the parameterized DBO is evaluated against a benchmark problem. Details of the test results and analysis are provided in the Appendix A. Upon comparison, the parametrically optimized DBO exhibits superior accuracy and stability in identifying the global optimal solution.

In addition to DBO, algorithms such as GWO, SSA, WOA, and PSO were also tested. The detailed test procedures, results, and analyses are provided in the Appendix A of this paper. Among these, the tuned DBO consistently demonstrates the highest accuracy and stability in locating the global optimal solution. The DBO-optimized VMD shows high stability and precision, effectively mitigating significant errors commonly introduced by manual parameter selection. Table 3 outlines the DBO parameter selection process, where 125 sets of parameter values were tested. Following a comparative performance analysis, the optimal values for DBO parameters were identified. Subsequently, the experimental data decomposition process for DVMD was conducted using a well-defined parameter optimization range, minimizing errors from manual parameter selection.

By using envelope entropy as the fitness value, the DBO algorithm optimizes VMD to determine the optimal number of decomposition layers, k, and the optimal penalty factor. If k is too small, modal aliasing effects may occur. Otherwise, it will decompose excessively. When the value is too large, the bandwidth of the modal function becomes too narrow, ignoring effective information. Alternatively, if the bandwidth is too wide, excessive interference information is included. The optimization parameters and their range for DBO are detailed in Table 4.

The DBO-optimized VMD algorithm features parameters k = 7 and α = 1537. In this case, the value of k is deemed appropriate, effectively decomposing the data’s complexity through modal decomposition. Simultaneously, this approach is scientific, retaining most of the essential information from the original modes while minimizing the interference as much as possible.

The parameter combination (7, 1537) was selected as optimal for the VMD decomposition of the experimental data; following DBO optimization, the DVMD algorithm decomposed the historical experimental data into seven IMF components and one residual, arranged from the highest to lowest frequency. Figure 7 displays the DVMD decomposition of the air quality data, while Figure 8 and Figure 9 illustrate the time-domain and frequency-domain plots of the decomposed IMF components.

Approximate entropy (ApEn) is a model-free statistic used to evaluate the complexity of time series, effectively capturing the characteristics of small fluctuations and nonlinear changes. For each IMF component sequence derived from VMD decomposition, a separate ApEn value is calculated, reflecting the component’s complexity. For a BIMF component, a high ApEn value indicates strong randomness and chaotic characteristics, including many tiny fluctuations and nonlinear elements; conversely, a low ApEn value suggests the component is simpler, more regular, and periodically characterized. This establishes a foundation for further analysis and processing of various components. The approximate entropy for each IMF component is calculated and presented in Table 5.

Based on the approximate entropy values from the table above, IMF1–IMF4 are classified as low-frequency component groups, and IMF5–IMF7 as high-frequency component groups. The low-frequency component group preserves the original air quality data, effectively eliminates random noise, and exhibits distinct time series characteristics. Given that the Informer exhibits superior generalization performance for extended time series, it processes the low-frequency component group of IMF1–IMF4. The high-frequency component group resembles Gaussian white noise in its data characteristics. CNN-LSTM is employed to predict IMF5–IMF7, owing to its enhanced predictive capabilities for high-frequency sequences.

### 3.5. Numerical Experiment Setup

To facilitate visual comparison of different air quality prediction models, this paper devises three sets of comparison experiments to validate the effectiveness of the DVMD-Informer-CNN-LSTM model.

Experimental Group 1 comprises models pertinent to the proposed model along with related benchmark models; Experimental Group 2 consists of models introduced in this paper and those central to mainstream research. To prove the advantages of the DBO optimization algorithm, Experimental Group 3 utilized optimization algorithms such as GWO, SSA, PSO, and WOA to substitute DBO and calculated the prediction outcomes after substituting these models to compare with the proposed model, thereby highlighting the performance benefits of DBO as used in the new model, as detailed in Table 6.

The evaluation metrics chosen for the experiment were R-squared ($R^{2}$), Root-Mean-Square Error (RMSE), and Mean Absolute Error (MAE). The formulas for the three metrics are as follows.
(23)$$R^{2}=1-\frac{\overset{i=1}{\underset{n}{\sum^{\;}}}{\left({\hat{y}}_{i}-y_{i}\right)}^{2}}{\overset{i=1}{\underset{n}{\sum^{\;}}}{\left(\frac{1}{n}\overset{i=1}{\underset{n}{\sum^{\;}}}y_{i}-y_{i}\right)}^{2}}$$
(24)$$RMSE=\sqrt{\frac{1}{n}\overset{i=1}{\underset{n}{\sum^{\;}}}{\left({\hat{y}}_{i}-y_{i}\right)}^{2}}$$
(25)$$MAE=\sqrt{\frac{1}{n}\overset{i=1}{\underset{n}{\sum^{\;}}}\left|{\hat{y}}_{i}-y_{i}\right|}$$

### 3.6. Numerical Experiment Results

#### 3.6.1. Experiment I

In this experiment, we performed ablation experiments. Predicted data plots were compared with raw data, revealing that the expected and experimental results for each model closely followed the observed trends. This indicates that the experimental setup for each group generated relatively accurate predictive data. Variations in air quality prediction performance among models are attributable to their differing prediction accuracies. Figure 10, Experiment I, shows more significant fluctuations in model prediction performance, particularly in the intervals [30, 50] and [110, 130]. The new model closely matches the observed values. The proposed model matches the observations very well, both being higher than the comparative models of the ablation experiments, and the Informer model has a poorer prediction performance. Compared with the newly proposed coupled model, the three benchmark models in the control group show lower prediction accuracies, with variance nodes deviating from the original observations.

#### 3.6.2. Experiment II

In this experiment, three classical and optimization algorithms and their combined models are used to compare with DVMD-Informer-CNN-LSTM. The classical algorithms comparison models include LSTM, GRU, Bi-LSTM, and Bi-GRU. The optimization algorithms and their combination models include GWO-VMD-Informer-CNN-LSTM, PSO-VMD-Informer-CNN-LSTM, SSA-VMD-Informer-CNN-LSTM, and WOA-VMD-Informer-CNN-LSTM four models.

The use of intelligent algorithm optimization, combined with data preprocessing and dual processing methods, has proven to be more effective in enhancing prediction accuracy than a single-strategy approach. Experiment II more clearly demonstrates the differences in prediction performance among models. As illustrated in Figure 11, the new model aligns closely with observed values, whereas the traditional neural network model underperforms in this interval. Among the four models in the comparison group, none achieve the prediction accuracy of the newly proposed coupled model for air quality forecasting. This further illustrates that DVMD is well-suited for nonlinear and non-stationary time series. The technique effectively eliminates pattern aliasing and yields decomposition results that are more conducive to accurate predictions. In intelligent algorithm experiments, the difference in prediction performance occurs in the interval [30, 50]. Figure 12, Experiment III, illustrates that the DVMD-optimized parameter model aligns closest with the predicted values, outperforming the GWO, SSA, PSO, and WOA optimization algorithms, with the control groups demonstrating lower prediction accuracies compared with the new coupled models. The prediction results across model groups display broadly similar trends. DBO exhibits faster and more robust search capabilities, making it particularly suitable for parameter optimization in the VMD model. DVMD enhances the generalization abilities of the coupled prediction model, thereby improving prediction accuracy. The new coupled model’s prediction results are closest to the original observations, showcasing its superior performance in air quality prediction.

Additionally, a comprehensive comparative analysis of the predicted values by the DVMD-Informer-CNN-LSTM model is depicted in Figure 13. Visual inspection of the comparison plots reveals the differences between the new model and the original data, effectively highlighting the model’s superior performance. This analysis not only confirms the new model’s accuracy but also underscores its effectiveness in managing complex air quality data, further validating its application in practical environmental monitoring scenarios.

In the figure, the gray area represents residual post-model prediction, the blue solid line denotes actual PM2.5 data, and the red dashed line illustrates the new model’s prediction results. The trends of both curves closely align, indicating a high degree of similarity. Residual fluctuations do not exceed 10, with 95% of errors within the range of [−5, 5]. Comparison results demonstrate that the new model provides higher prediction accuracy, smaller residuals between predictions and original observations and that the hybrid model effectively enhances prediction accuracy.

To illustrate the performance of the proposed DVMD-Informer-CNN-LSTM air quality prediction model, the results are quantitatively analyzed using the selected metrics: R-Square, MAE, and RMSE. The outcomes of the three sets of comparative experiments are presented in Table 7, Table 8 and Table 9.

In Experiment Group 1, our model was compared to the benchmark model proposed in this study to validate the new model. The prediction accuracy of the new model surpasses that of the benchmark model. Compared with the VMD-Informer-CNN-LSTM model without DBO optimization, the *MAE* decreased by 19.57% to 4.7733, RMSE by 20.15% to 2.1848, and $R^{2}$ improved by 3.5% to 0.9704. Data preprocessing that decomposes high-frequency and low-frequency sequences enhances the model’s predictive capabilities. The new model significantly outperforms both the Informer and CNN-LSTM models, with improvements including a 10.08% increase in $R^{2}$, a 29.03% reduction in RMSE, and a 49.64% decrease in MAE compared with the Informer model. Compared with the CNN-LSTM model, the $R^{2}$ improved by 3.99%, RMSE decreased by 77.85%, and *MAE* by 35.82%. Compared to the benchmark model, the proposed new model demonstrates higher prediction accuracy.

In Experiment Group 2, the new model is compared to mainstream single models to showcase its competitiveness. Among the four proposed new prediction models, the Bi-GRU model performs the best with the highest value of $R^{2}$ and the lowest RMSE and MAE ($R^{2}$ = 0.9291, RMSE = 9.9632, MAE = 6.7337). However, there is still a gap compared to the proposed new hybrid model. The new model, compared with the Bi-GRU model, shows an $R^{2}$ improvement of 3.99%, a 78.84% reduction in RMSE, and a 29.11% decrease in MAE. This indicates that the combined prediction model’s accuracy surpasses that of single models like the Bi-GRU.

In Experiment Group 3, the superiority of the DBO algorithm is demonstrated over other swarm intelligence optimization algorithms. Experiments using four different optimization strategies show that the PSO optimization algorithm performs the best among the proposed algorithms, with $R^{2}$ = 0.9565, RMSE = 2.3502, and MAE = 5.5237. The hybrid model, optimized with DBO parameters, is compared to the PSO optimization algorithm. The $R^{2}$ improved by 1.39%, RMSE decreased by 7.03%, and MAE by 13.59%, illustrating DBO’s superior stability and search capabilities over other optimization algorithms and its better suitability for optimizing VMD model parameters. Moreover, the DVMD is particularly suitable for nonlinear and nonsmooth time series, proving the DVMD-Informer-CNN-LSTM model’s effectiveness and superiority. The model’s prediction curve more closely matches actual values and exhibits higher accuracy than the PSO-optimized model.

## 4. Discussion

### 4.1. Conclusions

To enhance the accuracy of AQI prediction, given the complexity and non-stationarity of air quality time series, this paper integrates several existing optimization models and proposes a DVMD-Informer-CNN-LSTM-based predictive optimization model. Initially, the correlation coefficient method is employed to identify key influencing factors, integrating meteorological data with major pollutant types affecting air quality. Subsequently, the optimal number of decomposition modes and penalty parameters are determined using the DBO algorithm. The preprocessed data are then decomposed into subsequences and residuals using DVMD to reduce the complexity of the original data. The decomposed high-frequency and low-frequency data are predicted using the Informer and CNN-LSTM algorithms for the individual components derived from the DVMD decomposition. The prediction results are then combined to obtain the final predicted values. To evaluate the performance of the optimized DBO algorithm, comparison models such as the GWO, PSO, SSA, and WOA algorithms were selected, along with 9 benchmark functions and 18 problems from a total of 57 real optimization problems (Appendix A). This extensive testing verifies the superior performance of the DBO algorithm and the effectiveness of the proposed model. The benchmark test functions include four single-mode test functions (F1–F3 and F7), three multi-mode test functions (F8–F10), and two complex mode test functions (F16 and F18). Additionally, these 18 questions validate the excellent performance of the DBO algorithm and the predictive accuracy and rationality of the proposed model.

The model employs $R^{2}$, *RMSE*, and *MAE* as evaluation metrics. Comparison with the prediction results of related models demonstrates that the proposed new model exhibits high prediction accuracy. The main conclusions are as follows.

The DVMD-Informer-CNN-LSTM model outperforms the following models in predictive accuracy: VMD-Informer-CNN-LSTM, Informer, CNN-LSTM, Bi-LSTM, Bi-GRU, GRU, LSTM, GWO-VMD-Informer-CNN-LSTM, PSO-VMD-Informer-CNN-LSTM, SSA-VMD-Informer-CNN-LSTM, and WOA-VMD-Informer-CNN-LSTM models in prediction. Among them, the DVMD-Informer-CNN-LSTM model achieves the highest predictive accuracy (MAE = 4.7733, RMSE = 2.1848, $R^{2}$ = 0.9704), significantly enhancing predictive performance. Compared with other swarm intelligence optimization algorithms, the DBO algorithm effectively mitigates the issue of local optima, thereby enhancing its competitiveness. The parameter-optimized VMD is more suitable for long-term sequence feature extraction, demonstrating superior processing capability for nonlinear and nonsmooth sequences, thus enhancing the model’s predictive accuracy and generalization ability.

In this paper, we present an enhanced DVMD-Informer-CNN-LSTM hybrid optimization prediction model that significantly improves air quality prediction accuracy, contributing to advancements in the field. By integrating data smoothing, anomaly screening, and intelligent algorithm parameter optimization techniques, our approach reduces data complexity, minimizes errors from manual parameter tuning, enhances robust stability and generalization ability, and significantly improves prediction accuracy. The model captures both the intrinsic characteristics of the dataset and the influencing factors, ensuring consistent accuracy across different prediction objects and demonstrating robust applicability for predicting primary air pollutants.

### 4.2. Limitations

Although the proposed DVMD-Informer-CNN-LSTM model performs well in air quality prediction, it still has some limitations.

The integration of multiple algorithms increases the computational complexity. Although this integration improves the prediction accuracy, it also requires more computational resources and time. In future research, we will explore the dynamic integration method based on Bayesian optimization to further improve prediction accuracy [56].

In terms of dataset limitations, the original dataset only contains data from 13 air quality monitoring stations in Beijing. This regional limitation of the dataset may affect the generalization ability of the model. To enhance the generalization of the model, future research should be extended to data from different regions.

In terms of multi-factor integration, the current model is mainly based on air quality monitoring data for prediction. Future studies should consider integrating multiple factors, such as weather and geographic location information, to improve the accuracy and reliability of the prediction models in general. These limitations have become the focus of our future research to further enhance the performance and application scope of the proposed methods.

## Figures and Tables

**Figure 1 entropy-26-00534-f001:**
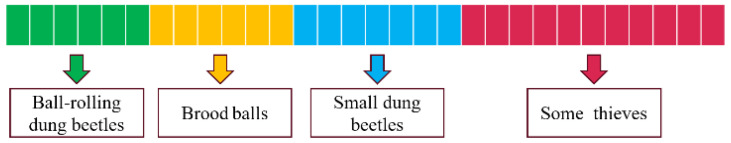
Distribution of search agents in the DBO algorithm.

**Figure 2 entropy-26-00534-f002:**
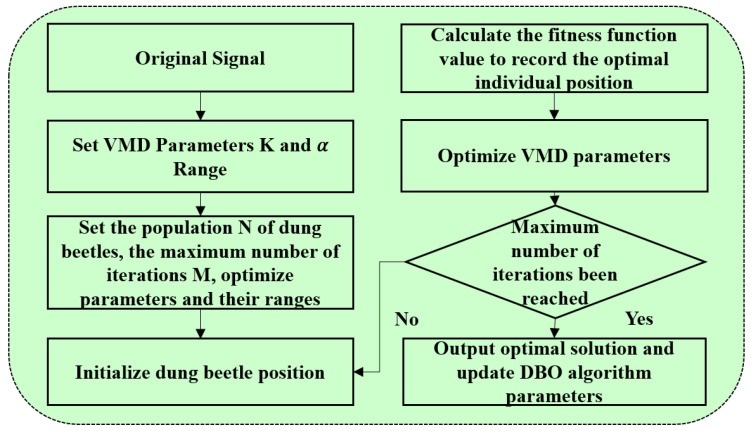
Flowchart for optimization of DVMD parameters.

**Figure 3 entropy-26-00534-f003:**
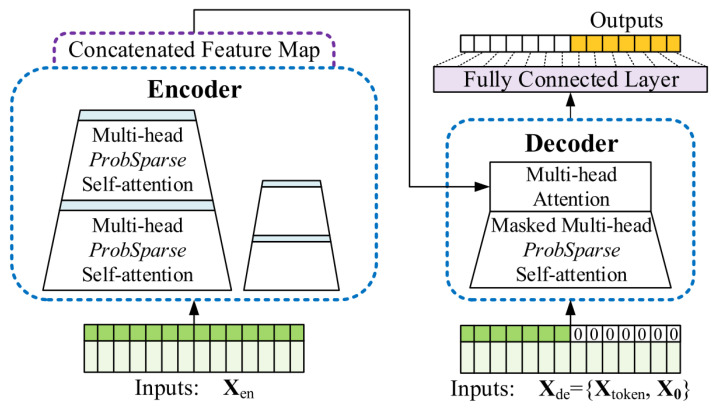
Principle of Informer encoder-decoder.

**Figure 4 entropy-26-00534-f004:**
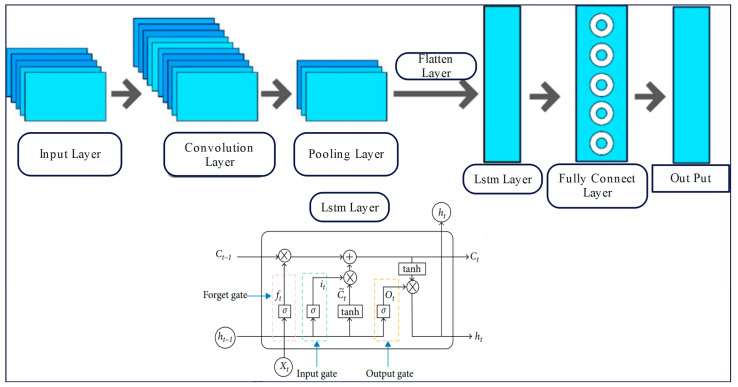
CNN-LSTM model structure.

**Figure 5 entropy-26-00534-f005:**
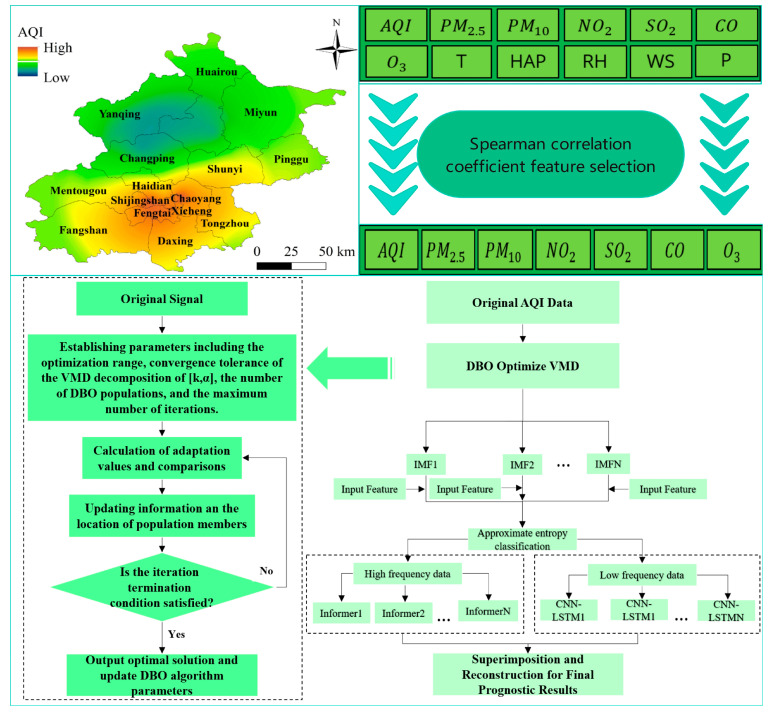
Flowchart of DVMD-Informer-CNN-LSTM combination model.

**Figure 6 entropy-26-00534-f006:**
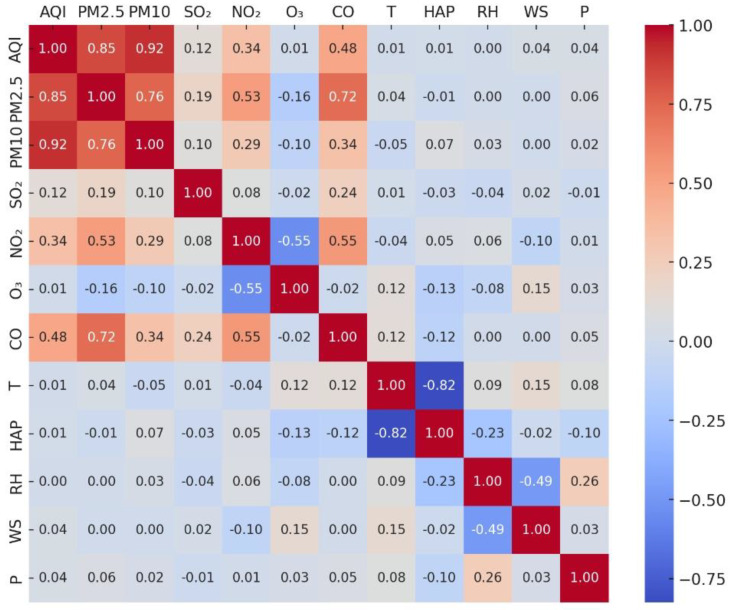
Feature correlation coefficient heat map.

**Figure 7 entropy-26-00534-f007:**
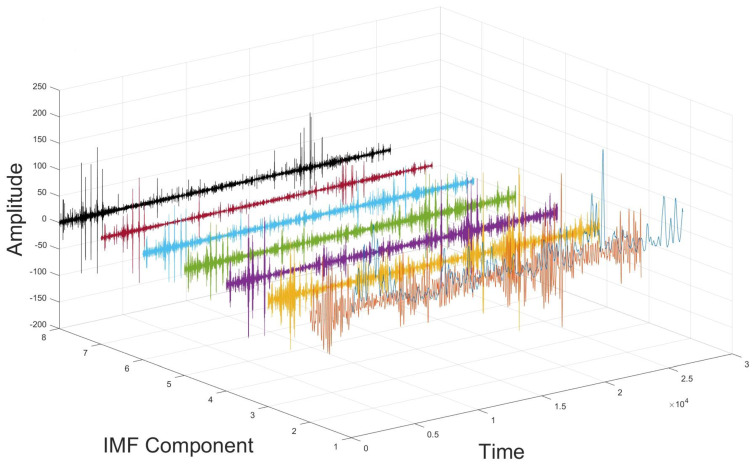
DVMD decomposition of air quality data.

**Figure 8 entropy-26-00534-f008:**
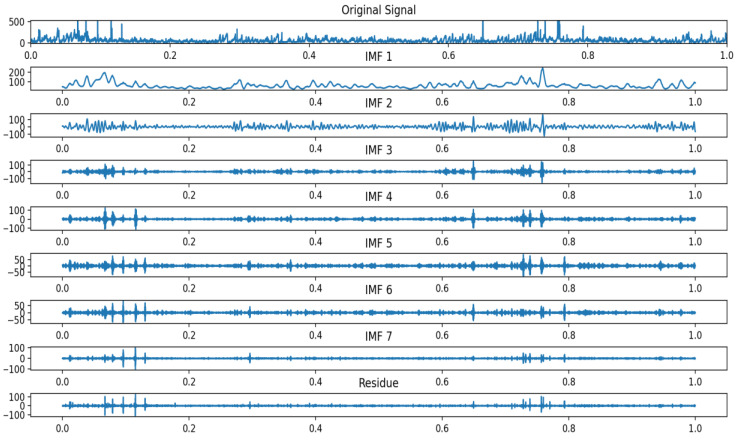
Decomposed IMF component time-domain plot.

**Figure 9 entropy-26-00534-f009:**
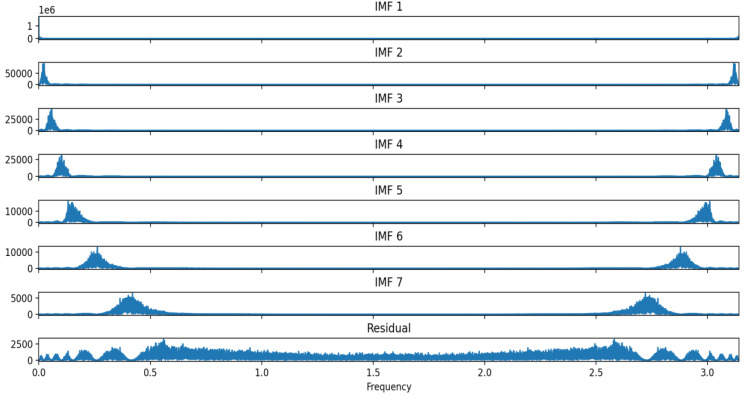
Decomposed IMF component frequency-domain plot.

**Figure 10 entropy-26-00534-f010:**
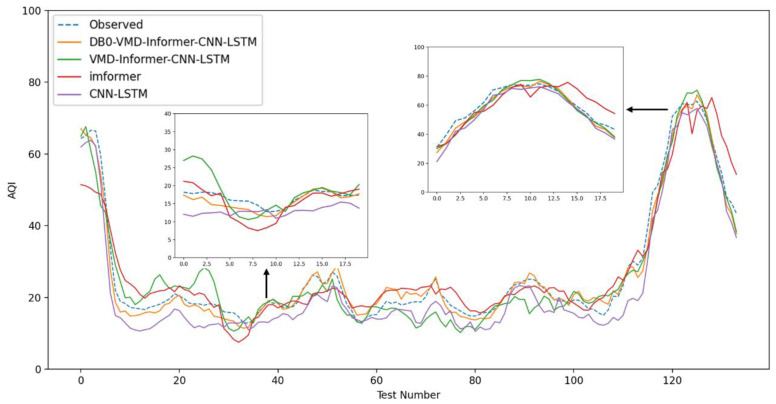
Model comparison figure number 1.

**Figure 11 entropy-26-00534-f011:**
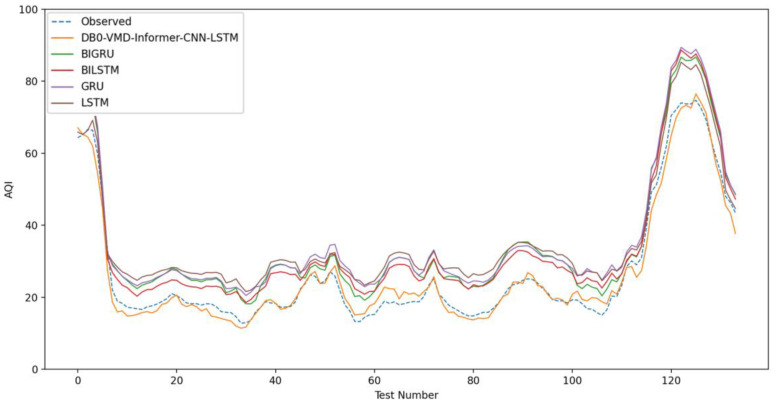
Model comparison figure number 2.

**Figure 12 entropy-26-00534-f012:**
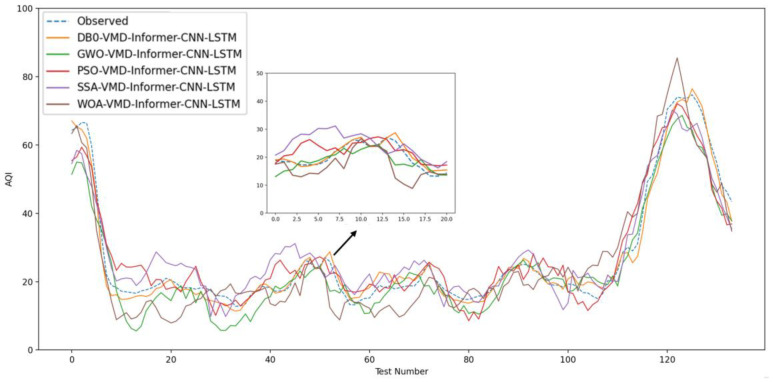
Model comparison figure number 3.

**Figure 13 entropy-26-00534-f013:**
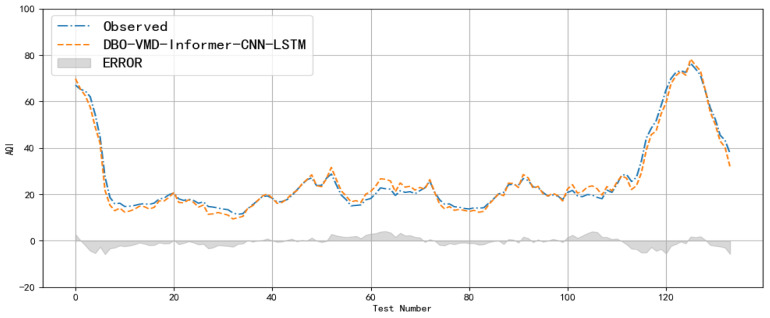
Residual plots for model comparison.

**Table 1 entropy-26-00534-t001:** Data Statistical tests.

Test	Test Statistic	Critical Value (5%)	*p*-Value	Implication
ADF	−29.542	−2.862	5.40 × 10^−28^	Stationary
KPSS	0.001	0.463	0.1	Stationary
Ljung-Box Q	4951.281	1.96	0.0	Autocorrelation

**Table 2 entropy-26-00534-t002:** Symbol description of indicators.

Symbol	Meaning	Unit
AQI	Air quality index	Dimensionless
PM2.5	Pulmonary particulate matter	μg/m^3^
PM10	Particulate matter	μg/m^3^
NO_2_	Nitrogen dioxide	μg/m^3^
CO	Nitric oxide	mg/m^3^
SO_2_	Sulfur dioxide	μg/m^3^
O_3_	Ozone	mg/m^3^
T	Temperature	°C
HAP	Horizontal atmospheric pressure	Pa
RH	Relative humidity	RH
WS	Wind speed	m/s
P	Precipitation	mm

**Table 3 entropy-26-00534-t003:** DBO parameter selection.

Parameter	Range	Optimum
k	0.05, 0.1, 0.15, 0.02	0.15
b	0.05, 0.25, 0.45, 0.65, 0.85	0.25
S	0.1, 0.5, 1, 1.5, 2	0.5
n	-	50
t	-	500

**Table 4 entropy-26-00534-t004:** DBO optimization of VMD parameters.

Parameter	Range	Optimum
K	2–10	7
α	500–2000	1537

**Table 5 entropy-26-00534-t005:** Approximate entropy of IMF components.

Component	Approximate Entropy
IMF1	0.020429335
IMF2	0.114359496
IMF3	0.217794642
IMF4	0.336460907
IMF5	0.510052784
IMF6	0.603696447
RES (IMF7)	0.752094373

**Table 6 entropy-26-00534-t006:** Prediction models of experiment group.

Model Group	Predictive Model	Model Introduction
Group 1	VMD-Informer-CNN-LSTM	The model proposed in this paper does not include optimization of DBO parameters.
	Informer [22]	Long’s proposed predictive model for indoor air quality captures temporal dependencies and correlations between data metrics.
	CNN-LSTM [18]	Yan’s proposed hybrid forecasting model is based on multi-temporal seasonal sites in Beijing.
Group 2	Bi-GRU [17]	Zhang uses a spatiotemporal depth algorithm to make multi-step predictions over a 24-h period.
	Bi-LSTM [16]	Multi-density layer Bidirectional Long Short Term Memory Neural Network for predicting PM2.5 concentration in indoor environment proposed by Prihatno.
	GRU [15]	Huang conducted a predictive assessment of the quality of the atmospheric environment using a dataset containing meteorological characteristics and air quality.
	LSTM [14]	Seng multi-indicator integrated air quality prediction model for predicting particulate matter concentration data at selected representative air quality sites in Beijing.
Group 3	GWO-VMD-Informer-CNN-LSTM	Parameter optimization of VMD by GWO.
	SSA-VMD-Informer-CNN-LSTM	Parameter optimization of VMD by SSA.
	PSO-VMD-Informer-CNN-LSTM	Parameter optimization of VMD by PSO.
	WOA-VMD-Informer-CNN-LSTM	Parameter optimization of VMD by WOA.

**Table 7 entropy-26-00534-t007:** Prediction models of Experiment Group 1.

Predictive Model	MAE	RMSE	R^2^
DBO-VMD-Informer-CNN-LSTM	4.7733	2.1848	0.9704
VMD-Informer-CNN-LSTM	6.9986	2.7569	0.9337
Informer	9.4788	3.0787	0.8696
CNN-LSTM	7.4373	9.8658	0.9305

**Table 8 entropy-26-00534-t008:** Prediction models of Experiment Group 2.

Predictive Model	MAE	RMSE	R^2^
DBO-VMD-Informer-CNN-LSTM	4.7733	2.1848	0.9704
Bi-LSTM	6.7525	10.3353	0.9277
Bi-GRU	6.7337	9.9632	0.9291
LSTM	7.8369	10.3180	0.9240
GRU	7.9263	10.9633	0.9218

**Table 9 entropy-26-00534-t009:** Prediction models of Experiment Group 3.

Predictive Model	MAE	RMSE	R^2^
DBO-VMD-Informer-CNN-LSTM	4.7733	2.1848	0.9704
GWO-VMD-Informer-CNN-LSTM	6.0013	2.4497	0.9517
SSA-VMD-Informer-CNN-LSTM	6.7259	2.5934	0.9476
PSO-VMD-Informer-CNN-LSTM	5.5237	2.3502	0.9565
WOA-VMD-Informer-CNN-LSTM	6.6898	2.5864	0.9406

## Data Availability

The data that support the findings of this study are available upon request from the authors.

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
