# Peer review of "Enhanced Air Quality Prediction Using a Coupled DVMD Informer-CNN-LSTM Model Optimized with Dung Beetle Algorithm"

_entropy, 2024, doi:10.3390/e26070534_

Round 1
Reviewer 1 Report
Comments and Suggestions for Authors The manuscript deals with enhancing air quality prediction Using a Coupled DVMD Informer-CNN-LSTM Model optimized with Dung Beetle algorithm. It is well-structured and easy to follow. However, the following comments should be addressed in order to be considered for publication. Comment. My major point is about the numerical experiments and the proposed methodology - How the SoA forecasting models were selected? Why models such as N-Beats, DeepTime, Sequence2Sequence, Autoformer, etc were not included? - The stationary property consists of primary importance for the effectiveness of a forecasting model. I highly recommend the authors to examine if the series satisfy the stationarity property. Along this line, the authors should use the framework presented in [A novel forecasting strategy for improving the performance of deep learning models. Expert Systems with Applications] and apply Augmented Dickey-Fuller (ADF) and Kwiatkowski-Phillips-Schmidt-Shin (KPSS) tests for exploring if the series are stationary or not. - The reliability of the evaluated models' forecasts must be evaluated by examining the existence of autocorrelation in the residuals. I recommend the authors to apply Ljung-Box Q-test for examining the hypothesis "the data exhibits no autocorrelation for a fixed number of lags L" (eg L = 10) [A novel multi-step forecasting strategy for enhancing deep learning models’ performance. Neural Computing and Applications, 34(22), 19453-19470] - A statistical analysis should be applied for making useful conclusions from the results of Tables 6-8 Comment. The presentation of the manuscript should be improved in order to follow the journal's standards - In Section 4, the authors should present the limitations of the proposed approach - Figures 2, 5, 6 should be in a "academic" style - too many colors - Figures 8 and 9 should be enlarged - The authors should double check the manuscript for addressing some typos.Comments on the Quality of English Language
Minor editing of English language required
Reviewer 2 Report
Comments and Suggestions for Authors
The authors propose a hybrid ensemble learning based on VMD and neural networks for forecasting. There are some major concerns:
1. Why do authors implement Informer for high-frequency data?
2. Why do authors choose VMD instead of DWT, EMD, and EWT? These decomposition techniques have shown success in forecasting.
3. How do the authors conduct decomposition? Improper decomposition leads to data leakage. The literature has shown how to address the data leakage problems. Time series forecasting based on echo state network and empirical wavelet transform.
4. How do the authors implement the ensemble? Is it static or dynamic? The dynamic ensemble usually shows better accuracy. Bayesian optimization based dynamic ensemble for time series forecasting.
5. The major weakness is the comparative study. Authors should compare to decomposition-based forecasting methods, such as VMDLSTM, EWTRVFL, EWTESN, EWTedRVFL.
6. Ablation study and sensitivity analysis are helpful.
Round 2
Reviewer 1 Report
Comments and Suggestions for Authors
My recommendation is that the manuscript can be accepted in present form. The author have properly addressed the previous comments.
Comments on the Quality of English LanguageMy recommendation is that the manuscript can be accepted in present form. The author have properly addressed the previous comments.